# Trampoline Stiffness Estimation by Using Robotic System for Quantitative Evaluation of Jumping Exercises

**DOI:** 10.3390/s23249645

**Published:** 2023-12-06

**Authors:** Gunseok Park, Seung-Hwan Choi, Chang-Hyun Kim, Min Young Kim, Suwoong Lee

**Affiliations:** 1Advanced Mechatronics Research Group, Daegyeong Division, Korea Institute of Industrial Technology, Daegu 42994, Republic of Korea; rjstjr010@kitech.re.kr (G.P.); csw1496@kitech.re.kr (S.-H.C.); limition@kitech.re.kr (C.-H.K.); 2School of Electronics Engineering, Kyungpook National University, Daegu 41566, Republic of Korea; 3Research Center for Neurosurgical Robotic System, Daegu 41566, Republic of Korea

**Keywords:** trampoline, robot manipulation, elastic constant and force estimation, Hooke’s law, linear regression

## Abstract

Trampolines are recognized as a valuable tool in exercise and rehabilitation due to their unique properties like elasticity, rebound force, low-impact exercise, and enhancement of posture, balance, and cardiopulmonary function. To quantitatively assess the effects of trampoline exercises, it is essential to estimate factors such as stiffness, elements influencing jump dynamics, and user safety. Previous studies assessing trampoline characteristics had limitations in performing repetitive experiments at various locations on the trampoline. Therefore, this research introduces a robotic system equipped with foot-shaped jigs to evaluate trampoline stiffness and quantitatively measure exercise effects. This system, through automated, repetitive movements at various locations on the trampoline, accurately measures the elastic coefficient and vertical forces. The robot maneuvers based on the coordinates of the trampoline, as determined by its torque and position sensors. The force sensor measures data related to the force exerted, along with the vertical force data at X, Y, and Z coordinates. The model’s accuracy was evaluated using linear regression based on Hooke’s Law, with Mean Absolute Error (MAE), Root Mean Square Error (RMSE), and Correlation Coefficient Squared (R-squared) metrics. In the analysis including only the distance between X and the foot-shaped jigs, the average MAE, RMSE, and R-squared values were 17.9702, 21.7226, and 0.9840, respectively. Notably, expanding the model to include distances in X, Y, and between the foot-shaped jigs resulted in a decrease in MAE to 15.7347, RMSE to 18.8226, and an increase in R-squared to 0.9854. The integrated model, including distances in X, Y, and between the foot-shaped jigs, showed improved predictive capability with lower MAE and RMSE and higher R-squared, indicating its effectiveness in more accurately predicting trampoline dynamics, vital in fitness and rehabilitation fields.

## 1. Introduction

Trampoline exercises are renowned for their significant benefits in enhancing lower limb muscular strength, physical fitness, and rehabilitation [1,2,3]. Prior research indicates that individuals of diverse age groups and health conditions can readily participate, with energy expenditure varying based on specific bouncing styles. This exercise is notably effective in improving physiological indicators [4,5,6,7]. The current trend in utilizing research and technology to maximize the health benefits of trampoline exercises highlights the importance of sensor-based quantitative evaluation tools.

Actions performed on the trampoline are categorized based on acceleration data obtained from a smartwatch [8]. A sensor attached to the trampolinist’s back measures jumping actions in coordinates, subsequently categorized into various jump types employing machine learning techniques [9]. Periodic G-force loads experienced by trampoline users under various conditions are measured through a characterization analysis of trampoline bounce using acceleration data [10]. Trampoline motion is categorized by attaching an inertial sensor to the trampolinist’s body [11]. In trampoline competitions, the correlation between the flight time of straight jumps and the balance of skeletal muscle mass and occlusal balance is analyzed [12]. Employing a three-axis accelerometer and gyroscope sensor device on the trampolinist, various dynamic conditions are determined [13]. By comparing the acceleration characteristics of three different trampoline models, differences in maximum acceleration and jerk based on these models are confirmed [14]. Table 1 outlines the methodological approaches utilized by various systems for the sensor-based quantitative evaluation of trampolines, along with the main limitations each system encounters.

Sensor-based quantitative assessment systems for trampoline use have limitations, such as the need to attach sensors to the user’s body or install them on the trampoline. In previous research, we utilized a deep-learning-based image-processing algorithm to estimate the three-dimensional position of the user’s feet on the trampoline using shadow images of the feet [15,16]. However, estimating the physiological indicators of trampoline exercise based on the shadow images of the user’s feet proved challenging. Since we estimated the depth coordinate of the feet using shadow images, we suggest that if the elastic modulus of the trampoline can be estimated, it would enable quantitative evaluation of trampoline exercises, such as jump power and calorie consumption. Therefore, estimating the trampoline’s elastic modulus is essential for a comprehensive evaluation of trampoline exercise.

The mechanical and kinetic energy characteristics of the double bounce phenomenon are analyzed based on a trampoline dynamic model, incorporating factors like stiffness, damping, air resistance, and friction [17]. The study assesses the control of lower muscle stiffness and the impact of mechanical energy processes during drop jumps on a sprung surface [18]. It also examines the impact on leg stiffness and the subjective experience during jumping on the trampoline’s elastic surface [19]. A mathematical model is established by estimating the trampoline user’s body mass and inertial characteristics while measuring the elasticity and damping characteristics of the trampoline surface [20]. The study evaluates the acute effects of trampoline training sessions on leg stiffness and reactive power, simultaneously exploring correlations with the participants’ gender [21]. Notably, both children and adults maintain their coordination structure while jumping on the mini trampoline, but children demonstrate an increase in vertical body stiffness to compensate for the reduced surface stiffness [22]. The investigation delves into the physical determinants of maximum flight time on the trampoline [23]. Trampoline safety and performance are evaluated by dropping weights onto different trampolines, measuring dynamic parameters [24]. Table 2 provides a detailed overview of the methodological approaches employed in each study focused on the elastic mechanics of trampolines, alongside a discussion of the primary limitations encountered in these studies.

Other studies of trampoline elasticity have struggled to derive accurate estimates due to limitations in the methodology used to estimate the modulus of elasticity. The common limitations of these studies on trampoline elasticity are methodological constraints such as narrow focus, limited generalizability, short-term analysis, and small sample size. Therefore, it was essential to measure the stiffness of trampolines through an objective and advanced method. To quantitatively evaluate trampoline motion, this study introduced a robotic system equipped with a foot-shaped jig to precisely measure vertical forces for different jumping motions at different locations on the trampoline. The system performs automatic repetitive motions to facilitate accurate and numerous data collection. In Section 2, we estimate the elastic constants and forces received by the robotic system through a linear regression model based on the vertical force data as a function of trampoline depth. Section 3 details the analysis of the performance metrics of the linear regression model, along with an analysis of the actual versus predicted values. Section 4 describes the proposed methodology for estimating the elastic constants and forces of the trampoline. Finally, Section 5 presents the conclusions.

## 2. Methods

### 2.1. Experiment Diagram

Figure 1 diagram depicts the system of interaction for control and data acquisition and in robot manipulation. Central to the system is a personal computer (PC) that conducts two-way communications with a robot controller. Movement commands sent from the PC to the robot controller are deciphered and implemented by a KUKA iiwa LBR 14 R820 robot. Affixed to the robot’s end-effector are foot-shaped jigs. Torque and position sensors, installed on the robot’s joints, control the movements, and force sensors attached to the end-effector amass vertical force data exerted on the robot. The collected force data are subsequently transmitted back to the PC for analytics. The force data are analyzed using Linear Regression according to Hooke’s Law to estimate the forces applied to the trampoline.

### 2.2. Experimental Environment and Equipment Description

Figure 2 shows an experimental setup with a robot set up next to a trampoline for data acquisition with a jig. The jig is attached to the end-effector of this robot manipulator. The trampoline used is a Jumping Corporation ‘J6H130 FLEXI’ model, with dimensions of 1360 mm × 1360 mm × 285 mm and a maximum load of 130 kg [25]. Jumping Corporation’s trampoline “J6H130 FLEXI” is made of high-quality steel circular profile with a wall thickness of 2 mm, which offers excellent strength and rigidity, making it durable and safe. Its polygonal shape and light weight of less than 13 kg increase its flexibility and ease of movement. The trampoline’s load-bearing structure features a unique hexagonal tubular steel skeleton, elastic rubber ropes for the fixing strings and six support legs. Characteristics such as material composition, surface strength, and structural design are important in estimating the elastic modulus of a trampoline. The material of the trampoline plays an important role in estimating the modulus of elasticity. Trampolines made of rubber or synthetic fibers can have greater elasticity than other materials. The surface strength of the trampoline determines its response to the forces applied by the robot. A stronger surface can withstand more force. The frame and support structure of the trampoline affects how the forces are distributed and transmitted. A stable and rigid structure allows for a more uniform distribution of forces, which helps with more accurate modulus of elasticity estimation. 

In the conducted experiment, the KUKA LBR iiwa 14 R820 model served as the robotic manipulator. This multifunctional and accurate robotic limb boasts a payload of 14 kg, demonstrating its capacity for carrying substantial weight. With a reach extending to 820 mm and encompassing 7 axes, it provides adaptable automation across varied positions. The robot’s precision is highlighted by its ability to repeat movements with a fine margin of 0.1 mm. Its weight, around 30 kg, makes it compatible with diverse mounting setups [26]. Equipped with a suite of sensors, including torque and position sensors, accelerometers, gyroscopes, and mechanisms for collision detection, the robot demonstrates a comprehensive sensory capacity. Each joint incorporates torque sensors to quantify forces and moments, and position sensors to ascertain angles and locations. Situated at the robot’s end-effector, the force sensor assesses linear forces, capturing both pressure applied to objects and tensile forces. This plethora of data is meticulously recorded at a rapid frequency of 1000 Hz. The robot’s manipulator end-effector comes fitted with a “Media flange Touch pneumatic” interface. A specific load, shaped like a foot, was affixed to this flange to facilitate the KUKA load identification routine, a process in which the robot autonomously calculates the weight, center, and inertial characteristics of the attached tool, seamlessly integrating these calculations into its motion control algorithms.

In Figure 3, the CAD model depicted in Figure 3a serves as a linkage between the CAD model of Figure 3b and the foot-shaped jig shown in Figure 3c, allowing for angular adjustments of the jig. The CAD model in Figure 3b facilitates the connection between the CAD models of Figure 3a,d, ensuring that the foot-shaped jig can be mounted onto the manipulator. The CAD model in Figure 3c features foot-shaped jig sizes ranging from 210 mm to 300 mm at 10 mm intervals, based on maximum length, for both left and right feet. The CAD model in Figure 3d is designed to adjust the distance between the dual foot-shaped jigs, with a predetermined spacing of 50 mm between the large circular slots. The jigs are fabricated from SUS (Stainless Steel).

### 2.3. Manipulator Coordinate System Configuration and Movement Range Analysis

Before initiating manipulator movement, it is imperative to establish the coordinate system through Robot Programming. The robot’s position can be identified in Figure 4a. The initial position of the manipulator is set to the center of the trampoline with (X, Y) coordinates (0, 0), and the point where the trampoline mat meets the manipulator is set as the starting point for the Z coordinate. The X direction of the manipulator is horizontal to the robot platform, indicating forward and backward movement. +X points forward, while −X points backward. Similarly, the Y direction is horizontal to the robot platform, indicating left and right movement. +Y signifies left, and −Y denotes right. The Z direction is vertical to the robot platform, indicating upward and downward movement. +Z points upwards, while −Z points downwards (vertical direction of the trampoline’s elastic surface). Since the distance the manipulator moves is measured in mm, the unit will be omitted in subsequent coordinate data descriptions. In Figure 4b, the yellow-marked area illustrates the two-dimensional (X, Y) maximum workspace by the manipulator on the trampoline. Given the trampoline’s symmetry, the vertex coordinates of this maximum area are A (0, 225), B (0, −225), C (−250, 125), and D (−250, −125), which accounts for half of the trampoline. Depending on the distance between the two foot-shaped jigs, Y coordinates range from +Y to −Y, X coordinates from 0 to −X, and Z coordinates from 0 to −Z. Data on three-dimensional (X, Y, Z) coordinates and the applied force at each point on the trampoline were collected. The end-effector’s workspace permits movement in the +X direction to a certain extent but cannot reach the maximum +X point of the trampoline mat. The maximum torque for Z-direction movement of the robot diminishes as it moves in the +X direction, and the trampoline’s elasticity imposes restrictions on pressing beyond a certain depth in the Z direction.

### 2.4. Analysis of Manipulator Movement Range

The distance between the dual foot-shaped jigs is adjusted using the function presented in Figure 3d in 100 mm intervals based on the center of the end-effector of the manipulator. Data were collected by installing jigs of various foot shapes for each distance setting. In Figure 5, the blue areas of Figure 5a–e indicate the (X, Y) coordinate range when the distance of the two foot-shaped jigs is 100, 200, 300, 400, and 500 mm, respectively. All coordinates in (X, Y) decrease at intervals of 25 mm, and at each (X, Y) position, along the Z-axis, they decrease from 0 to −100 mm at intervals of 5 mm. The distance between the dual foot-shaped jigs is referred to as “FD” and the foot size is referred to as “FS”. As the FD value increases, the two-dimensional (X, Y) movable area on the trampoline by the manipulator reduces. With the FD at 100 mm in Figure 5a, the movement in Y-axis was 450 mm at X coordinate of 0 and the movement in Y-axis was 250 mm at X coordinate −175 mm. In Figure 5b, where the FD was 200 mm, the Y-axis movement was 400 mm at X coordinate 0 and the Y-axis movement was 200 mm at X coordinate −175 mm. In Figure 5c, where the FD was 300 mm, the Y-axis movement was 300 mm at X coordinate 0 and Y-axis movement was 100 mm at X coordinate −175 mm. In Figure 5d, where the FD was 400 mm, the Y-axis movement was 200 mm at X coordinate 0 and Y-axis movement was 50 mm at X coordinate −175 mm. In Figure 5e, with a FD of 500 mm, the Y-axis movement was 100 mm at X coordinate 0 and Y-axis movement was 50 mm at X coordinate −100 mm. Table 3 summarizes the manipulator movement range coordinate information by FD.

### 2.5. Data Acquisition and Distribution

An external computer running a Python client application was employed to communicate in real-time with the KUKA iiwa robot’s controller. The experiment’s focus was on using the force sensor located in the robot’s manipulator end-effector. This sensor played a crucial role in pushing against a trampoline and accurately measuring the vertical forces applied. For programming and controlling the KUKA robot, the KUKA Sunrise Workbench-1.16.2.16 software was utilized. The robot’s movement sequence was carefully programmed. It started from a pre-set HomePosition, moving to a specified targetXY location. This movement was executed using point-to-point (PTP) motion, calibrated at a relative joint speed of 0.5, equivalent to 50% of the robot’s maximum speed. Upon reaching the targetXY coordinates, the robot was then directed to proceed to the targetZ position. This phase of the movement was accomplished through linear motion, executed at a Cartesian speed of 500 mm/s. The distinction between joint speed and Cartesian speed in this context is significant. Joint speed refers to the relative speed of each individual joint in the robot’s structure, while Cartesian speed relates to the speed at which the robot’s tool end, or the end-effector, moves. The robot then paused for 5 s to collect force data at that location, and repeated the rest of the way, changing the X, Y, and Z coordinates.

The data collected at varying FDs yield the following results. At a FD of 100 mm, 3466 data points for X coordinates, Y coordinates, Z coordinates, and force data were gathered, totaling 34,660 data points. Similarly, at 200 mm, 3004 data points were collected, amounting to a total of 30,040 data points. For 300 mm, 2080 data points were obtained, resulting in a total of 20,800 data points. At 400 mm, 1450 data points were gathered, with a total of 14,500 data points. Finally, at 500 mm, 610 data points were acquired, totaling 6100 data points. Altogether, 106,100 data points encompassing X coordinates, Y coordinates, Z coordinates, and force data were compiled. To enhance clarity regarding the correlation between the Z value and N, the depth values along the Z-axis were converted from negative to positive and stored. Figure 6 presents a two-dimensional distribution based on the force values at (Y, X) coordinates with respect to Z. Each (Y, X) coordinate group is distinctively colored using a random color scheme. The X-axis, representing Z values, is marked at intervals of 5 mm up to a total of 100 mm. The Y-axis, depicting Force values, is scaled at intervals of 100 N, extending up to 800 N. 

Figure 7 shows a three-dimensional representation of the X, Y, Z coordinate data, colored according to Force values. Higher Force values are represented in shades of yellow, while lower values are depicted in shades of blue. The X-axis, indicating X values, is marked at intervals of 50 mm up to a maximum of −250 mm. The Y-axis, representing Y values, is scaled from −200 mm in 100 mm intervals up to 200 mm. The Z-axis, displaying Z values, is marked at intervals of 20 mm, covering a total range of 100 mm.

### 2.6. Trampoline Elastic Response: Linear Regression Analysis Using the Extended Hooke’s Law

The elasticity of the trampoline constitutes a crucial variable in understanding the relationship between its depth and the vertical force it generates. This elasticity governs the trampoline’s ability to swiftly return to its original state and its capacity to absorb and release energy. Achieving height on the trampoline necessitates an adequate depth and force. When a user jumps onto the trampoline, it absorbs energy, causing it to depress, and subsequently releases this energy during recovery to its initial state. This released energy propels the user to a higher position. In this context, accurately comprehending the relationship between the trampoline’s depth and the generated vertical force is of paramount importance. To analyze this relationship, Hooke’s law was applied.

Hooke’s law, a fundamental principle of physics, elucidates the linear relationship between the deformation of an elastic object and the resulting restoring force. It underscores the direct relationship between the force generated when extending or compressing an object and the amount of deformation. This relationship is expressed in the following equation
(1)F=−k×∆x,
where *F* refers to the restoring force, *k* is the elastic constant, and ∆*x* is the amount of deformation.

The applicability of this principle extends to the case of the trampoline. Given that Hooke’s law embodies a linear relationship and bears similarity to a linear regression model, it becomes plausible to perform linear regression analysis. When an individual jumps on the trampoline or applies pressure to it, the trampoline’s surface deforms. The extent of deformation directly corresponds to the generation of a restoring force, facilitating the trampoline’s return to its initial state. To forecast the trampoline’s response using Hooke’s law, an extended model that accounts for several additional features is necessary. The extension of Hooke’s law enables a more accurate prediction and description of the trampoline’s complex response.

In the application of the extended Hooke’s law, linear regression analysis was executed concerning the vertical force received in correspondence to the trampoline’s depth. This analysis encompassed various combinations of features, including X, Y, FD, and FS. The dataset was partitioned into a training dataset and a test dataset, at an 80:20 ratio. The features were grouped into a total of eight combinations: X and FS, X and FD, Y and FS, Y and FD, X and Y, X, Y, and FS, X, Y, and FD, and X, Y, FD, and FS. The elastic modulus and force of the trampoline in relation to the vertical force received based on the depth of the trampoline were estimated for each feature group. Figure 8 showcases the linear regression lines for X and FS in Figure 8a, X and FD in Figure 8b, Y and FS in Figure 8c, Y and FD in Figure 8d, X and Y in Figure 8e, X, Y, and FS in Figure 8f, X, Y, and FD in Figure 8g, and X, Y, FD, and FS in Figure 8h. We also use a color map to provide different colors for each group.

## 3. Results

In Figure 9, the model’s performance is evaluated through a visual comparison of actual measured force data and predicted force data. Specifically, Figure 9 depicts the distribution of predicted values in relation to the actual values for various feature combinations: X and FS in Figure 9a, X and FD in Figure 9b, Y and FS in Figure 9c, Y and FD in Figure 9d, X and Y in Figure 9e, X, Y, and FS in Figure 9f, X, Y, and FD in Figure 9g, and X, Y, FD, and FS in Figure 9h. Notably, the model’s performance is deemed better when the predicted values closely converge to the y = x line. A visual assessment of model performance reveals that Figure 9b,d,g,h exhibit exceptional performance. Notably, these groups share the common feature of FD.

The evaluation of the linear regression model’s performance in estimating the elastic constant and force of the trampoline focused on four exceptional cases among the various feature combination groups in Figure 9. To assess performance, three evaluation indicators were selected: Mean Absolute Error (MAE), Root Mean Squared Error (RMSE), and Correlation Coefficient Squared (R-squared). From MAE, RMSE, and R-squared, we found Min, Max, Q1 (First Quartile), Q3 (Third Quartile), median, and mean. Q1 is the median of the lower half of the dataset. It represents the 25th percentile, which means 25% of the data points are below this value. Q3 is the median of the upper half of the dataset. It represents the 75th percentile, meaning 75% of the data points are below this value. The range from Q1 to Q3 is called the interquartile range (IQR). A wide IQR range means that the data are spread out. This indicates that the data are inconsistent and vary a lot. It can be affected by extreme values or outliers. On the other hand, a narrow IQR range means that the data as a whole tends to be stable and consistent. In this way, Q1 and Q3 can help you understand the central tendency and variability of your data, and whether outliers are present.

MAE is defined by Equation (2), where *y_i_* is the actual value, $\hat{y_{i}}$ is the predicted value, and n is the observed value. MAE is the mean value of the sum of the absolute difference between the actual value and predicted value, representing the average error size of the predicted value. This indicator is robust against outliers due to its consideration of the absolute values of individual error. Thus, the MAE loss function does not distinguish between large and small errors, implying that the prediction model approaches the actual observed value as the MAE value becomes smaller.
(2)$$\mathrm{MAE}=\frac{1}{n}\sum_{i=1}^{n}\left|y_{i}-\hat{y_{i}}\right|$$

In the boxplot of MAE, lower values indicate smaller errors. As listed in Table 4, the group with features X, Y, FD and FS has a higher Max value compared to other groups, and as illustrated in Figure 10, several outlier values are present. With the other three groups, the groups with features X and FD, and Y and FD have a larger Median and Mean value compared to the group with features X, Y and FD. The Q3-Q1 values of the groups with features X, FD and X, Y and FD were 5.2958 and 5.7326, respectively, which showed a consistent level of predictability.

RMSE is defined as Equation (3) and is the square root of MSE. MSE is defined as Equation (4), where *yi* is the actual value, $\hat{y_{i}}$ is the predicted value, and n is the observed value. MSE is the mean value of the squared difference between the actual value and predicted value. RMSE allows the size of the prediction error to be interpreted in the same scale as the actual units. This evaluation indicator aids to interpret the size of the prediction error in units of the original data. Akin to MSE, this indicator responds sensitively to large errors and is sensitive to outliers. This indicates that the smaller the RMSE value, the better the model fits the data and the average error between the predicted value and the actual value is smaller. MSE and RMSE are related metrics, but they serve different purposes. MSE is valuable during model training, while RMSE is more suitable for reporting and understanding the model’s performance in the context of the original data.
(3)$$\mathrm{RMSE}=\sqrt{MSE}$$
(4)$$\mathrm{MSE}=\frac{1}{n}\sum_{i=1}^{n}\left(y_{i}-\hat{y_{i}}\right)$$

In the boxplot of RMSE, lower values indicate smaller errors. In Table 5, the group with features X, Y, FD and FS has a higher Max value compared to other groups, and in Figure 11, there are many outlier values. With the other three groups, the groups with features X and FD, and Y and FD have a larger Median and Mean value compared to the group with features X, Y and FD. The Q3-Q1 values of the groups with features X, FD and X, Y and FD were 5.2958 and 5.7326, respectively, which showed a consistent level of predictability.

R-Squared, as defined in Equation (5), offers insights into the model’s ability to explain data variability, with values ranging from 0 to 1. In the formula are two key components used to calculate the R-squared value, which is a statistical measure of how close the data are to the fitted regression line. ${SS}_{res}\;($Sum of Squares of Residuals) measures the amount of variance in the dependent variable that is not explained by the independent variable(s) in the model. In other words, it quantifies how much the data points deviate from the fitted line. Mathematically, it is the sum of the squares of the differences between the observed values and the values predicted by the model. ${SS}_{tot}($Total Sum of Squares) measures the total variance in the dependent variable. It is calculated as the sum of the squares of the differences between the observed values and their mean. Essentially, it quantifies how much the data points deviate from their mean value. A value closer to 1 indicates a strong capacity to elucidate data variability. While R-Squared is an intuitive indicator, it lacks information about the absolute size of errors. Consequently, higher R-squared values signify that the model closely fits the data, and the model’s predictions closely align with actual data.
(5)$$R^{2}=1-\frac{{SS}_{res}}{{SS}_{tot}}$$

In the context of the box plot for R-squared, higher values indicate a greater explanatory power of the model. The group with X, Y, FD, and FS features displays a lower minimum value when compared to other groups, and Figure 12 presents numerous outlier values. For the other three groups, median and mean values are distributed at similar levels. In Table 6, the Q3–Q1 values of the groups with features X, FD and X, Y and FD were 0.0029 and 0.0046, respectively, which were lower than those of other groups, indicating a higher level of consistency in predictability.

## 4. Discussion

In this study, we estimated the elastic constant of the trampoline and the force received using a robot equipped with two foot-shaped jigs. The foot-shaped jig was mounted in a single direction on the robot for data collection. This method does not encompass all possible movements of the foot; hence, it is not a complete solution. However, it offers the advantage of efficiently collecting a large amount of data in an automated manner, compared to previous studies. We established a linear regression model based on an extended Hooke’s law, which provides a reliable linear regression estimation within a certain depth range. However, it is important to note that in cases where the force applied to the trampoline exceeds a certain threshold, a force larger than what a linear addition would suggest may be required. This can lead to a shift towards nonlinear data distribution. While it is recognized that in such situations, the elastic constant of the trampoline and the force received should be estimated using a nonlinear regression model, the results of this study offer a simplified approach for estimating these parameters.

The experimental results reveal a substantial variation in the trampoline’s response, contingent upon the FD feature within the estimation model. Furthermore, when comparing Figure 9g,h, it becomes evident that the FS feature has a limited impact. In addition, based on the metrics of MAE, RMSE, and R-squared, the FS feature can be considered relatively inconsequential, despite the presence of outliers exhibiting minor error deviations. 

The determination of the elastic constant of the trampoline’s motion and the resulting vertical force carries significant implications for both manufacturers and users. From the manufacturer’s perspective, precise data pertaining to the elastic constant of the trampoline and the force generated serve as a pivotal reference during the product design and development phase. These data play a critical role in optimizing the performance and safety of the product. For the end-users, understanding the elastic constant of the trampoline and the associated force is paramount in ensuring a secure exercise environment. This knowledge empowers users to exert better control over their exercise techniques and allows them to harness the maximum benefits of trampoline exercise. The data collected in our study enable the identification of methods to enhance the effectiveness of trampoline workouts. By analyzing factors such as jump height or modifications in technique based on the elastic constant, training methods can be fine-tuned.

It is imperative to recognize that research into the elastic constant of trampolines and the resultant forces is a central factor in maximizing safety and efficacy within the realms of fitness and rehabilitation. Such knowledge equips users to experience a more secure and effective exercise environment, contributing to an overall improved exercise experience. In future research plans, a camera will be installed beneath a trampoline to capture video during the moments when a foot-shaped jig, mounted on a robot, compresses the trampoline. The aim is to develop a system that can estimate force using only the visual data collected from these recordings.

## 5. Conclusions

This study introduced a robotic system designed for the estimation of the elastic constant and the applied force on a trampoline. A specially designed robot, equipped with precision jigs, was positioned adjacent to the trampoline. These jigs, constructed from stainless steel, were intricately connected by several CAD parts. The robot’s manipulator’s range of movement, relative to the trampoline’s coordinates, was contingent upon the spacing between these foot-shaped jigs. A comprehensive dataset of 106,100 data points was collected for analysis.

To estimate the force applied to the trampoline, a linear regression model was developed, employing various combinations of input features. The model’s performance was systematically evaluated using well-established metrics such as MAE, RMSE, and R-Squared. Particularly noteworthy was the model’s performance when certain combinations included the “FD” feature. For the combinations involving X and FD, as well as X, Y, and FD, the three performance indicators demonstrated excellence. Specifically, the mean values for MAE, RMSE, and R-Squared for the X and FD combination were 17.9702, 21.7226, and 0.9840, respectively. In the case of X, Y, and FD, the mean values for MAE, RMSE, and R-Squared were 15.7347, 18.8226, and 0.9854, respectively.

For future investigations, there is a need to enhance prediction accuracy by minimizing the disparity between actual and predicted values. This can be achieved by implementing a nonlinear regression model to analyze the elastic constant of the trampoline and the vertical force it receives. For estimating the elasticity modulus of trampolines, the implemented robotic system facilitates automated collection of extensive data based on varying foot positions. This approach enhances estimation accuracy and enables quantification of trampoline exercise effects, such as calorie expenditure. Such an improvement not only reduces the risk of injury but also allows for the provision of personalized exercise programs to users through precise analysis. Consequently, the fields of home fitness, remote rehabilitation, and related industries are anticipated to witness substantial growth.

## Figures and Tables

**Figure 1 sensors-23-09645-f001:**
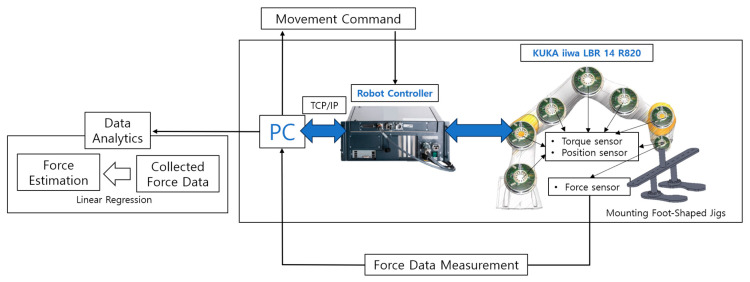
A robotic system for control and data measurement in manipulation tasks.

**Figure 2 sensors-23-09645-f002:**
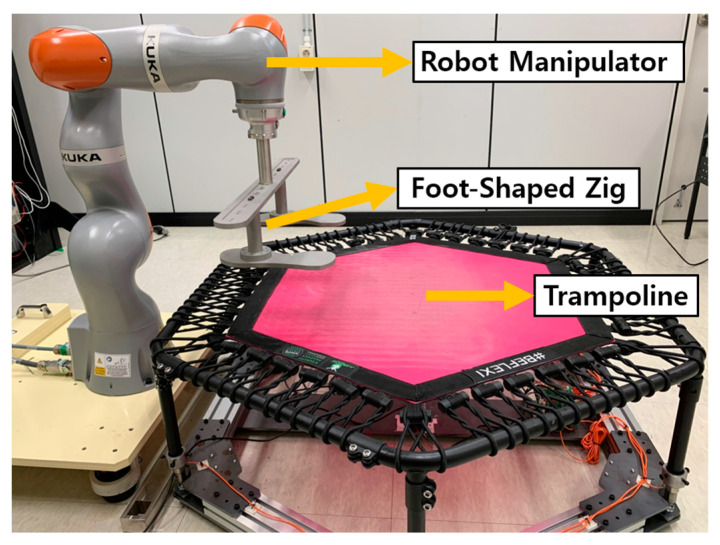
Experimental setup including Trampoline, Robot manipulator, and Foot-Shaped Jig.

**Figure 3 sensors-23-09645-f003:**
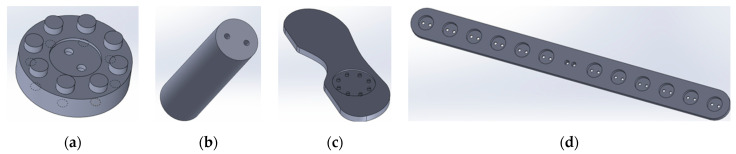
CAD configuration and functional description: (**a**) Angle adjuster; (**b**) Jig connector; (**c**) Foot-shaped jig; (**d**) Distance adjuster.

**Figure 4 sensors-23-09645-f004:**
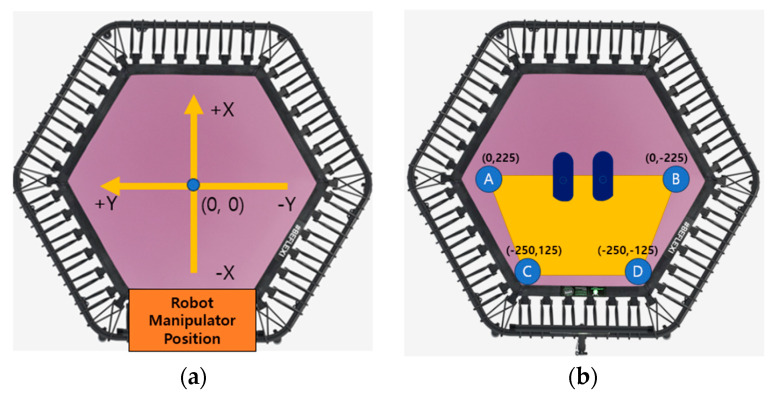
Manipulator movement and workspace configuration on the trampoline: (**a**) Robot coordinate system and initial position setup; (**b**) Indication of manipulator’s maximum working area.

**Figure 5 sensors-23-09645-f005:**
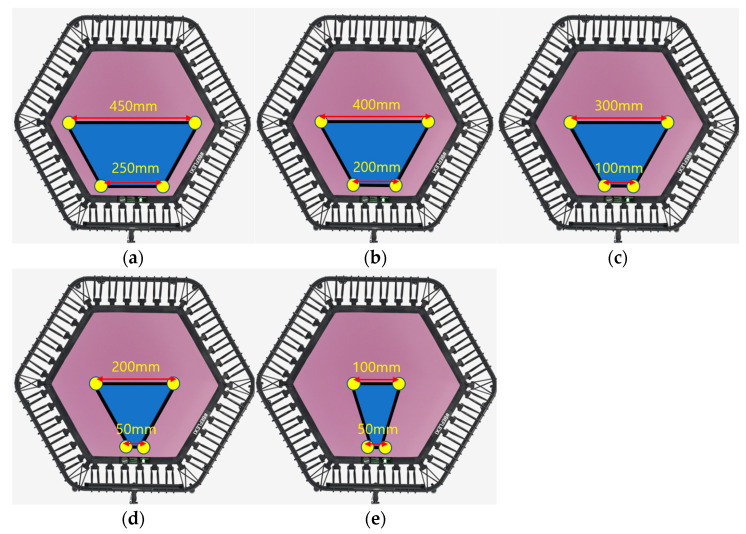
2D movement range of the manipulator based on jig spacing: (**a**) 100 mm; (**b**) 200 mm; (**c**) 300 mm; (**d**) 400 mm; (**e**) 500 mm.

**Figure 6 sensors-23-09645-f006:**
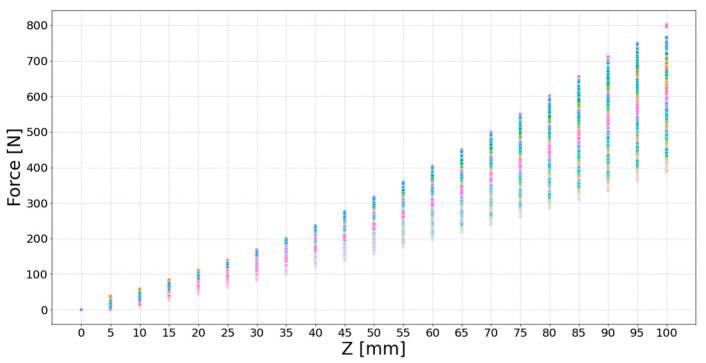
2D distribution of Force values based on Z for (Y, X) coordinates.

**Figure 7 sensors-23-09645-f007:**
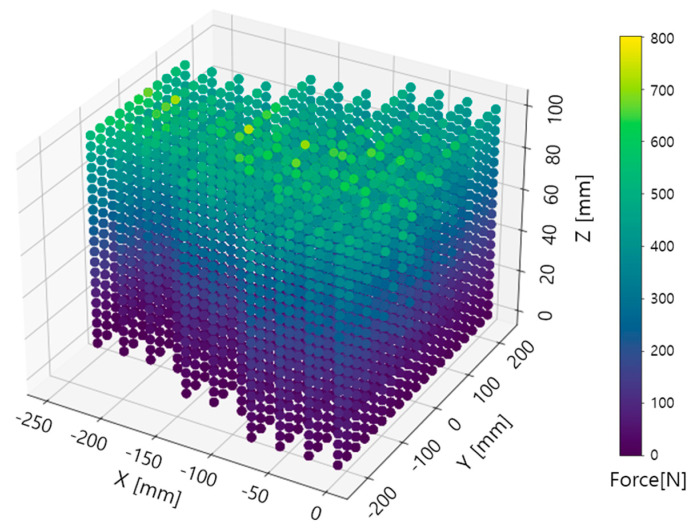
3D distribution of X, Y, and Z data with Force represented in color.

**Figure 8 sensors-23-09645-f008:**
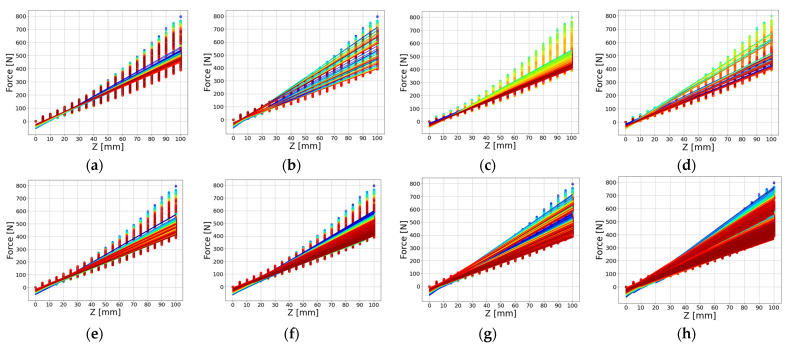
Linear Regression lines of force based on trampoline Z for different feature groups: (**a**) X and FS; (**b**) X and FD; (**c**) Y and FS; (**d**) Y and FD; (**e**) X and Y; (**f**) X, Y and FS; (**g**) X, Y and FD; (**h**) X, Y, FD and FS.

**Figure 9 sensors-23-09645-f009:**
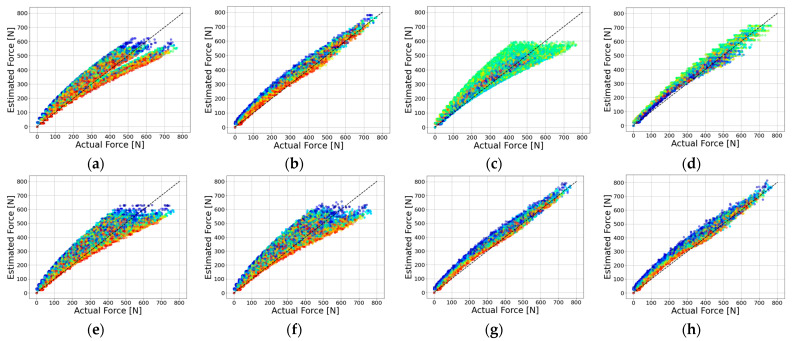
Actual force data and estimated force data comparison for different feature groups: (**a**) X and FS; (**b**) X and FD; (**c**) Y and FS; (**d**) Y and FD; (**e**) X and Y; (**f**) X, Y and FS; (**g**) X, Y and FD; (**h**) X, Y, FD and FS.

**Figure 10 sensors-23-09645-f010:**
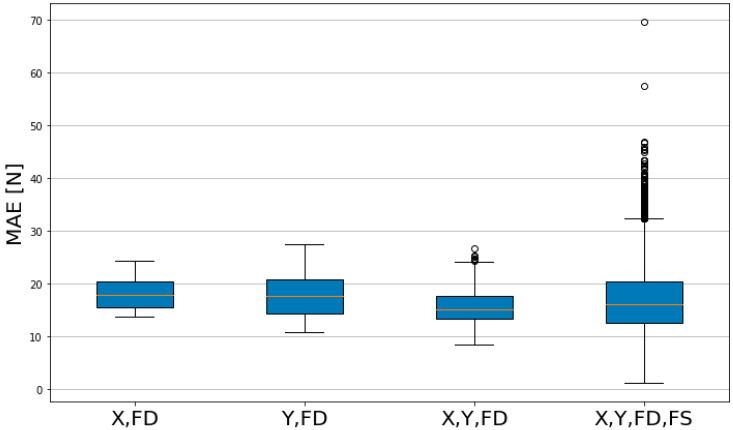
Comparison of MAE model performance by feature group.

**Figure 11 sensors-23-09645-f011:**
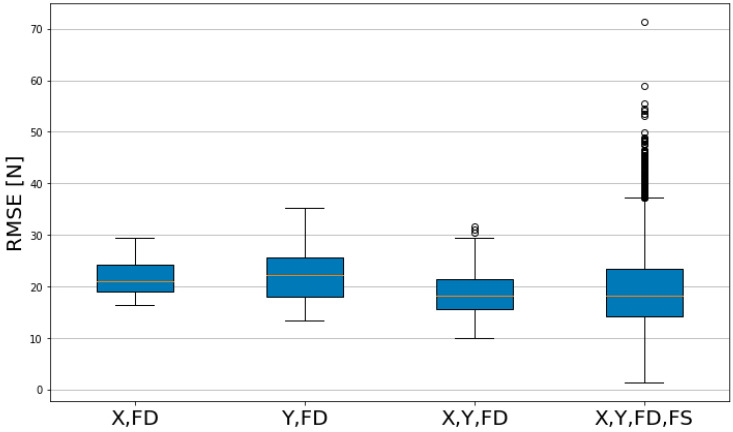
Comparison of RMSE model performance by feature group.

**Figure 12 sensors-23-09645-f012:**
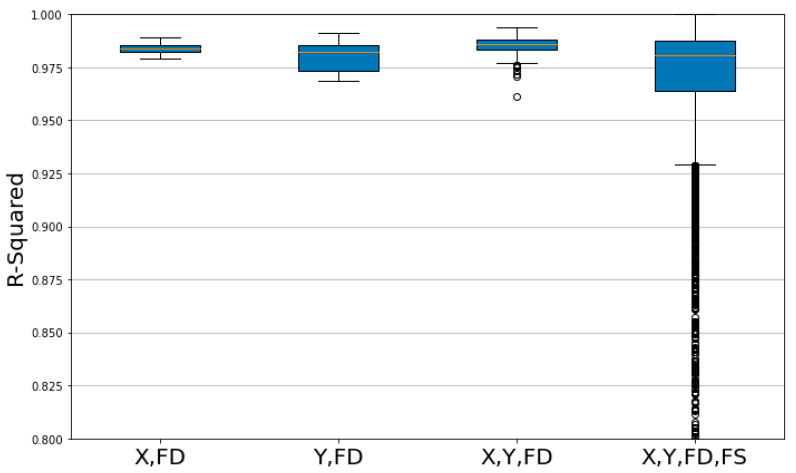
Comparison of R-Squared model performance by feature group.

**Table 1 sensors-23-09645-t001:** Sensor-based quantitative assessment system for trampoline.

System	Methodology/Limitation	
Smartwatch-based trampoline exercise monitoring [8].	Methodology	CNN with smartwatch data for trampoline motion detection.
	Limitation	Inaccurate in distinguishing complex motions.
Inertial Measurement Units (IMUs)- based jump detection and classification in trampoline gymnastics [9].	Methodology	Utilizes IMUs and machine learning for detailed classification of 50 trampoline jumps.
	Limitation	Challenged by reliance on high-quality sensor data for general applicability.
Tri-axial Accelerometer-based characterization of trampoline bounce using acceleration [10].	Methodology	Incorporates a tri-axial accelerometer and high-speed camera for trampoline biomechanics analysis.
	Limitation	Complex and equipment-specific, with a narrow biomechanical focus.
Inertial Sensor-based classification of trampoline jumps [11].	Methodology	Applies inertial sensors for automated trampoline jump classification.
	Limitation	Reliant on sensor quality and precise placement, affecting broad applicability.
Impact of muscle mass and dental occlusion on trampoline jump flight time [12].	Methodology	Assesses how muscle mass and dental occlusion affect trampoline jump flight time.
	Limitation	Narrow focus on internal biomechanical factors, limiting broader applicability.
Deep-learning-based image recognition for analyzing trampoline movements [13].	Methodology	Applies deep learning for image recognition to analyze trampoline movements from videos.
	Limitation	Relies heavily on video quality and computational power, with a narrow focus on trampoline movements.
Analyzing bounce characteristics on trampolines with a triaxial accelerometer [14].	Methodology	Uses a triaxial accelerometer to analyze bounce characteristics on different trampolines with experienced athletes.
	Limitation	Focuses on a small, skilled athlete group, limiting broader applicability.

**Table 2 sensors-23-09645-t002:** Studies in trampoline elasticity dynamics.

Studies	Methodology/Limitation	
Elastic Dynamics and Double Bounce Analysis in Trampoline Use [17].	Methodology	Employs simulations and experiments to study trampoline dynamics and the double bounce effect.
	Limitation	Concentrates solely on the double bounce, overlooking other aspects of trampoline dynamics.
Biomechanical Analysis of Leg Stiffness in Trampoline Jumping [18].	Methodology	Examines leg stiffness and energy dynamics in jumping using biomechanical analysis.
	Limitation	Lacks real-world applicability in trampoline jumping scenarios.
Motor and Sensory Adaptations in Trampoline Jumping Aftereffects [19].	Methodology	Examines motor and sensory changes after transitioning from trampoline to stiff surface jumping.
	Limitation	Primarily studies aftereffects, not ongoing trampoline use.
Elastic Dynamics of Somersaults on Trampolines [20].	Methodology	Models and analyzes the elastic dynamics of somersaults on a trampoline.
	Limitation	Specifically targets the elastic dynamics of somersaults, limiting broader application.
Assessing Acute Effects of Mini Trampoline Training on Leg Stiffness and Reactive Power in Adults [21].	Methodology	Assesses acute changes in leg stiffness and reactive power post-mini trampoline training.
	Limitation	Immediate, single-session effects without a control group.
A Comparative Study of Hopping Dynamics on Stiff and Elastic Surfaces in Children and Adults [22].	Methodology	Motion and force data were collected as participants hopped on two surfaces.
	Limitation	The fixed 1.5 Hz hopping frequency on the mini-trampoline may not reflect individual children’s preferences.
Exploring the Kinetics of Elite Trampolining: Physical Performance Measures and Jumping Time Analysis [23].	Methodology	Elite trampolinists’ jumping time of flight was analyzed using floor-based physical tests and countermovement jumps.
	Limitation	Small sample size and extrapolated load–velocity measures.
Impact Dynamics and Performance Analysis of Domestic Trampolines [24].	Methodology	Trampolines were evaluated by dropping varying weights from a set height.
	Limitation	Findings specific to tested trampolines, excluding spring and bed properties.

**Table 3 sensors-23-09645-t003:** Summary of manipulator movement coordinates based on jig spacing.

FD [mm]	Coordinate	Start [mm]	Finish [mm]
100	Y1	225	−225
	X1	0	−75
	Y2	175	−175
	X2	−100	−150
	Y3	125	−125
	X3	−175	−250
200	Y1	200	−200
	X1	0	−75
	Y2	150	−150
	X2	100	−150
	Y3	100	−100
	X3	−175	−250
300	Y1	150	−150
	X1	0	−75
	Y2	100	−100
	X2	−100	−150
	Y3	50	−50
	X3	−175	−250
400	Y1	100	−100
	X1	0	−75
	Y2	75	−75
	X2	−100	−150
	Y3	25	−25
	X3	−175	−250
500	Y1	50	−50
	X1	0	−75
	Y2	25	−25
	X2	−100	−150

**Table 4 sensors-23-09645-t004:** The box plot values and the mean value for MAE.

Feature Group Combination	MAE [N]					
	Min	Max	Q1	Q3	Median	Mean
X and FD	13.6902	24.2691	15.5118	20.3577	17.7471	17.9702
Y and FD	10.7176	27.4522	14.3071	20.6843	17.5435	17.7580
X, Y and FD	8.3368	26.5930	13.2603	17.6679	14.9969	15.7347
X, Y, FD and FS	1.1094	69.6129	12.5340	20.4346	16.0294	16.9649

**Table 5 sensors-23-09645-t005:** The box plot values and the mean value for RMSE.

Feature Group Combination	RMSE [N]					
	Min	Max	Q1	Q3	Median	Mean
X and FD	16.2817	29.3807	18.8800	24.1758	21.0593	21.7226
Y and FD	13.4311	35.1588	17.8992	25.6800	22.1676	22.3381
X, Y and FD	9.9059	31.5727	15.6367	21.3694	18.1531	18.8226
X, Y, FD and FS	1.2562	14.2552	14.2552	23.4618	18.2725	19.4996

**Table 6 sensors-23-09645-t006:** The box plot values and the mean value for R-Squared.

Feature Group Combination	R-Squared					
	Min	Max	Q1	Q3	Median	Mean
X and FD	0.9790	0.9890	0.9825	0.9854	0.9841	0.9840
Y and FD	0.9686	0.9910	0.9735	0.9856	0.9825	0.9800
X, Y and FD	0.0961	0.9938	0.9832	0.9879	0.9858	0.9854
X, Y, FD and FS	−427.0800	0.9999	0.9641	0.9877	0.9808	0.8225

## Data Availability

Data are contained within the article.

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
