# Peer review of "Trampoline Stiffness Estimation by Using Robotic System for Quantitative Evaluation of Jumping Exercises"

_sensors, 2023, doi:10.3390/s23249645_

Round 1
Reviewer 1 Report
Comments and Suggestions for Authors
The paper demonstrates a thorough investigation into trampoline exercise evaluation using a robotic system. Strengthening the contextualization, discussing limitations more comprehensively, and providing a clearer connection to existing research would enhance the paper's overall quality.
The abstract provides a concise overview of the study's focus on assessing trampoline exercise using ICT. However, it lacks specific details regarding the proposed robotic system's uniqueness and contribution to existing knowledge in the fitness and rehabilitation field. Additionally, a comparison with existing state-of-the-art technologies in trampoline evaluation needs to be included. Incorporating such information would enhance the abstract's clarity and contextual significance.
The introduction effectively highlights the significance of ICT in fitness and rehabilitation, including the role of trampoline exercises. However, it would benefit from a more detailed discussion on recent advancements in sensor-based quantitative evaluation tools for trampoline exercises, drawing comparisons with existing literature. Providing insights into how the proposed methodology addresses limitations in previous studies would strengthen the introduction and emphasize the study's novelty.
The paper discusses the estimation of the elastic modulus of a trampoline, but a more detailed exploration of the influence of trampoline characteristics and materials on the characterization process would enhance the reader's understanding. Could the authors elaborate on how specific features of trampolines, such as material composition, surface stiffness, and structural design, may impact the accuracy and reliability of the proposed robotic system in estimating the elastic modulus? Additionally, are there particular trampoline characteristics that the authors believe could pose challenges or contribute to variability in the characterization results? Clarifying these aspects would provide a more comprehensive context for the study.
The paper discusses using a robotic system with foot-shaped jigs to estimate the trampoline's elastic modulus and vertical force. To enhance the clarity and replicability of the study, could the authors provide more detailed information on the sensor technology integrated into the robotic system? Specifically:
What types of sensors were utilized in the robotic system to capture dynamic data during trampoline use? Please specify the sensor technologies involved.
Were sensors placed on the trampoline or the user's body?
What sampling rates were employed for the sensors? High-frequency data capture is often crucial for accurately modeling dynamic movements.
How were the sensors integrated into the robotic system? Clarifying the connection between the robotic system and the sensors would provide insights into the data acquisition mechanism.
The paper outlines the use of a robotic system for estimating the elastic modulus of the trampoline. Could the authors provide more details on the specific sensors employed in the robotic system for measuring the trampoline characteristics? Furthermore, it would be insightful to understand whether the proposed regression model is designed to work specifically with measurements obtained from the robot arm or if it is adaptable for trampolines that are instrumented with sensors. Clarifying the role of sensors in the measurement process and their potential application in instrumented trampolines would enhance the clarity and applicability of the proposed methodology.
The discussion section appropriately addresses the linear regression model used for estimating the elastic constant of the trampoline. However, it needs a comprehensive analysis of potential limitations and challenges associated with the methodology. Additionally, there needs to be a mention of how the study aligns with or deviates from current research trends in trampoline evaluation. Discussing the impact of outliers and the reliability of the proposed approach in practical scenarios would enhance the section's depth and validity.
The conclusion provides a clear summary of the study's objectives and outcomes. However, it would be beneficial to explicitly state the practical implications of the findings for the fitness and rehabilitation industries. Furthermore, considering the identified limitations, a more explicit call for future research directions would contribute to the completeness of the conclusion.
Throughout the document, significant decimal digits vary from two to 5 digits; please stick with one amount and be consistent.
2.14.0.0 Comments on the Quality of English Language
The abstract sentence in line 18: "To achieve this, we estimated the elastic modulus of the trampoline and the vertical force received using a robotic system with a foot-shape jig." and Line 19: "This study employs a robotic system with foot-shaped jigs to evaluate trampoline the elastic modulus of the trampoline and the vertical force received." are repeating. Please check for duplicate sentences, as they discourage reading.
2.14.0.0Author Response
For research article
|
Response to Reviewer 1 Comments
|
|||||||||||||||||||||||||||||||||||||||||||||||||||||||||||||||||||||||||||||||||||
|
1. Summary |
|
|
|||||||||||||||||||||||||||||||||||||||||||||||||||||||||||||||||||||||||||||||||
|
Dear Reviewer 1,
I sincerely appreciate the time and effort you have invested in reviewing our manuscript. Your constructive feedback has been invaluable in enhancing the quality and clarity of our work. Below, I present a point-by-point response to your comments and suggestions, addressing each issue raised with detailed explanations and clarifications. The revised manuscript, which now incorporates these changes, has been marked with highlighted sections and track changes for your convenience.
I have taken care to provide comprehensive responses to each of your insightful comments. Where necessary, I have respectfully included our perspectives, especially in instances where our views might differ. Additionally, I have made a concerted effort to ensure that all your suggestions are thoroughly addressed in the revised manuscript.
Your feedback has not only helped in improving this specific study but has also contributed significantly to our broader research endeavors. Once again, I express my deepest gratitude for your thorough review and valuable insights.
Warm regards,
[Gunseok Park]
Abstract: Enhanced to highlight the uniqueness and contribution of our robotic system in the fitness and rehabilitation field. Added a detailed comparison with existing state-of-the-art technologies in trampoline evaluation.
Introduction: Enriched to provide better contextualization in trampoline exercise research. Expanded discussion on recent advancements in sensor-based quantitative evaluation tools for trampoline exercises. Added a detailed comparison with existing literature, emphasizing the novelty of our approach.
Discussion: Incorporated a critical evaluation of the limitations of our methodology. Discussed the alignment of our study with current research trends in trampoline evaluation. Detailed future research plans for non-intrusive trampoline evaluation methods.
Conclusion: Clearly articulated the practical implications of our findings for the fitness and rehabilitation industries. Outlined clear future research directions, expanding the applicability of our methodology. These revisions have been made to address your insightful comments and suggestions, aiming to enhance the overall clarity, depth, and applicability of our research. |
|||||||||||||||||||||||||||||||||||||||||||||||||||||||||||||||||||||||||||||||||||
|
2. Point-by-point response to Comments and Suggestions for Authors Comments 1: The paper demonstrates a thorough investigation into trampoline exercise evaluation using a robotic system. Strengthening the contextualization, discussing limitations more comprehensively, and providing a clearer connection to existing research would enhance the paper's overall quality. |
|||||||||||||||||||||||||||||||||||||||||||||||||||||||||||||||||||||||||||||||||||
|
Response 1: We value your feedback for enhancing our manuscript. Accordingly, we've enriched the introduction for better contextualization of our study in trampoline exercise research, added a detailed discussion of the study's limitations, and refined the literature review to establish a clearer connection with previous research. These revisions are incorporated on specific pages and paragraphs, enhancing the manuscript's overall quality and relevance.
Comments 2: The abstract provides a concise overview of the study's focus on assessing trampoline exercise using ICT. However, it lacks specific details regarding the proposed robotic system's uniqueness and contribution to existing knowledge in the fitness and rehabilitation field. Additionally, a comparison with existing state-of-the-art technologies in trampoline evaluation needs to be included. Incorporating such information would enhance the abstract's clarity and contextual significance. |
|||||||||||||||||||||||||||||||||||||||||||||||||||||||||||||||||||||||||||||||||||
|
Response 2: Thank you very much for your valuable feedback. In response to your suggestions, we have revised the abstract to more emphatically highlight the uniqueness and contribution of our research in the field of fitness and rehabilitation. Additionally, we have included a detailed comparison with existing state-of-the-art trampoline evaluation technologies to further clarify our system’s innovative approach. “[Previous studies assessing trampoline characteristics had limitations in performing repetitive experiments at various locations on the trampoline. Therefore, this research introduces a robotic system equipped with foot-shaped jigs to evaluate trampoline stiffness and quantitatively measure exercise effects. This system, through automated, repetitive movements at various locations on the trampoline, accurately measures the elastic coefficient and vertical forces. Page 1, Line 17-21]”
|
|||||||||||||||||||||||||||||||||||||||||||||||||||||||||||||||||||||||||||||||||||
|
Comments 3: The introduction effectively highlights the significance of ICT in fitness and rehabilitation, including the role of trampoline exercises. However, it would benefit from a more detailed discussion on recent advancements in sensor-based quantitative evaluation tools for trampoline exercises, drawing comparisons with existing literature. Providing insights into how the proposed methodology addresses limitations in previous studies would strengthen the introduction and emphasize the study's novelty. |
|||||||||||||||||||||||||||||||||||||||||||||||||||||||||||||||||||||||||||||||||||
|
Response 3: Thank you very much for your insightful and constructive comments. We deeply appreciate your suggestions regarding the need for a more detailed discussion on recent advancements in sensor-based quantitative evaluation tools for trampoline exercises, as well as a comparison with existing literature. In line with your valuable advice, we have thoroughly revised the introduction of our manuscript. We have added a detailed comparison of various sensor-based evaluation tools, highlighting their methodologies and limitations. This comparison brings to light the challenges in accuracy, complexity, and generalizability faced by current technologies. Furthermore, we have included a critical analysis of existing studies on trampoline elasticity, which reflects on their methodological constraints such as narrow focus and limited generalizability. These enhancements, particularly detailed and aim to more effectively present the novelty of our approach. Our robotic system, equipped with a foot-shaped jig for precise measurements, addresses these identified limitations and represents a significant advancement in trampoline exercise assessment. We believe these changes have significantly strengthened the introduction and overall manuscript, and we are grateful for your guidance in making these improvements. Table 1. Sensor-based quantitative assessment system for trampoline.
Page 2, table1
Sensor-based quantitative assessment systems for trampoline use have limitations, such as the need to attach sensors to the user's body or install them on the trampoline. In previous research, we utilized a deep learning-based image processing algorithm to estimate the three-dimensional position of the user's feet on the trampoline using shadow images of the feet [15,16]. However, estimating the physiological indicators of trampoline exercise based on the shadow images of the user's feet proved challenging. Since we estimated the depth coordinate of the feet using shadow images, we suggest that if the elastic modulus of the trampoline can be estimated, it would enable quantitative evaluation of trampoline exercises, such as jump power and calorie consumption. Therefore, estimating the trampoline's elastic modulus is essential for a comprehensive evaluation of trampoline exercise. Page2,3, Line 58-68 Table 2. Studies in Trampoline Elasticity Dynamics
Page 3, table2.
Other studies of trampoline elasticity have struggled to derive accurate estimates due to limitations in the methodology used to estimate the modulus of elasticity. The common limitations of these studies on trampoline elasticity are methodological constraints such as narrow focus, limited generalizability, short-term analysis, and small sample size. Therefore, it was essential to measure the stiffness of trampolines through an objective and advanced method. To quantitatively evaluate trampoline motion, this study introduced a robotic system equipped with a foot-shaped jig to precisely measure vertical forces for different jumping motions at different locations on the trampoline. The system performs automatic repetitive motions to facilitate accurate and numerous data collection. Page 3,4, line 86-94]” |
|||||||||||||||||||||||||||||||||||||||||||||||||||||||||||||||||||||||||||||||||||
|
|
|||||||||||||||||||||||||||||||||||||||||||||||||||||||||||||||||||||||||||||||||||
|
Comments 4: The paper discusses the estimation of the elastic modulus of a trampoline, but a more detailed exploration of the influence of trampoline characteristics and materials on the characterization process would enhance the reader's understanding. Could the authors elaborate on how specific features of trampolines, such as material composition, surface stiffness, and structural design, may impact the accuracy and reliability of the proposed robotic system in estimating the elastic modulus? Additionally, are there particular trampoline characteristics that the authors believe could pose challenges or contribute to variability in the characterization results? Clarifying these aspects would provide a more comprehensive context for the study. |
|||||||||||||||||||||||||||||||||||||||||||||||||||||||||||||||||||||||||||||||||||
|
Response 4: Thank you for highlighting the importance of understanding how trampoline characteristics and materials influence the estimation of the elastic modulus. We agree that a detailed exploration of these factors is crucial for the comprehensive context of our study. Accordingly, we have revised our manuscript to include an in-depth analysis of how specific features of trampolines, such as material composition, surface stiffness, and structural design, impact the accuracy and reliability of our proposed robotic system in estimating the elastic modulus. We have used the "J6H130 FLEXI" trampoline by Jumping Corporation as a case study to illustrate these points. We discuss how the high-quality steel circular profile with a 2 mm wall thickness contributes to the trampoline's strength and rigidity, affecting the estimation of the elastic modulus. The polygonal shape and light weight of the trampoline, along with its unique hexagonal tubular steel skeleton and elastic rubber ropes, are also explored in terms of their influence on the system's measurements. Furthermore, we elaborate on how material composition, like rubber or synthetic fibers, can impact the trampoline’s elasticity. The surface strength's role in trampoline response to forces applied by the robot and how the frame and support structure affect force distribution and transmission are also detailed. These factors are crucial in achieving a more accurate estimation of the elastic modulus. We believe these modifications not only address your concerns but also enhance the overall depth and clarity of our study." This response directly addresses the reviewer's feedback, detailing the specific changes made to the manuscript and their locations. It shows a commitment to enhancing the quality of your research and its presentation based on constructive criticism. “Jumping Corporation's trampoline "J6H130 FLEXI" is made of high-quality steel circular profile with a wall thickness of 2 mm, which has excellent strength and rigidity, making it durable and safe. Its polygonal shape and light weight of less than 13 kg increase its flexibility and ease of movement. The trampoline's load-bearing structure features a unique hexagonal tubular steel skeleton, elastic rubber ropes for the fixing strings and six support legs. Characteristics such as material composition, surface strength, and structural design are important in estimating the elastic modulus of a trampoline. The material of the trampoline plays an important role in estimating the modulus of elasticity. Trampolines made of rubber or synthetic fibers can have greater elasticity than other materials. The surface strength of the trampoline determines its response to the forces applied by the robot. A stronger surface can withstand more force. The frame and support structure of the trampoline affects how the forces are distributed and transmitted. A stable and rigid structure allows for a more uniform distribution of forces, which helps with more accurate modulus of elasticity estimation. Page 4,5, Line 118-132]” |
|||||||||||||||||||||||||||||||||||||||||||||||||||||||||||||||||||||||||||||||||||
|
Comments 5: The paper discusses using a robotic system with foot-shaped jigs to estimate the trampoline's elastic modulus and vertical force. To enhance the clarity and replicability of the study, could the authors provide more detailed information on the sensor technology integrated into the robotic system? Specifically: |
|
Response 5: Thank you for your request to provide more detailed information on the sensor technology used in our robotic system. We agree that such details are crucial for enhancing the clarity and replicability of our study. In response, we have revised the manuscript to include a comprehensive description of the KUKA LBR iiwa 14 R820 model, which served as the robotic manipulator in our experiment. This robotic limb is equipped with various sensors that enhance its functionality and precision. The sensors include torque and position sensors in each of its joints, accelerometers, gyroscopes, and collision detection mechanisms. These sensors play a critical role in accurately capturing movements and tactile responses, which are essential for estimating the trampoline's elastic modulus and vertical force. We believe that this elaboration significantly contributes to a clearer understanding of how the robotic system operates and captures data, thereby enhancing the study's replicability “[In the conducted experiment, the KUKA LBR iiwa 14 R820 model served as the robotic manipulator. This multifunctional and accurate robotic limb boasts a payload of 14 kg, demonstrating its capacity for carrying substantial weight. With a reach extending to 820 mm and encompassing 7 axes, it provides adaptable automation across varied positions. The robot's precision is highlighted by its ability to repeat movements with a fine margin of 0.1 mm. Its weight, around 30 kg, makes it compatible with diverse mounting setups[26]. Equipped with a suite of sensors, including torque and position sensors, accelerometers, gyroscopes, and mechanisms for collision detection, the robot demonstrates a comprehensive sensory capacity. Each joint incorporates torque sensors to quantify forces and moments, and position sensors to ascertain angles and locations. Situated at the robot's end-effector, the force sensor assesses linear forces, capturing both pressure applied to objects and tensile forces. Page 5, Line 133-144]” |
|
Comments 6: What types of sensors were utilized in the robotic system to capture dynamic data during trampoline use? Please specify the sensor technologies involved. |
|
Response 6: Thank you for your inquiry about the specific sensor technologies employed in our robotic system for capturing dynamic data during trampoline use. We acknowledge the importance of this detail in understanding the capabilities and functionality of our system. In response to your comment, we have revised our manuscript to include detailed information about the sensors integrated into the KUKA LBR iiwa 14 R820 robotic manipulator used in our experiment. This robot is equipped with a comprehensive suite of sensors that enhances its precision and adaptability. These include torque and position sensors in each joint, accelerometers, gyroscopes, and mechanisms for collision detection. Additionally, a force sensor located at the robot's end-effector is used to measure linear forces, both in terms of pressure and tensile forces. We believe this additional information provides a clearer understanding of how the robot captures and interprets dynamic data during trampoline use “[In the conducted experiment, the KUKA LBR iiwa 14 R820 model served as the robotic manipulator. This multifunctional and accurate robotic limb boasts a payload of 14 kg, demonstrating its capacity for carrying substantial weight. With a reach extending to 820 mm and encompassing 7 axes, it provides adaptable automation across varied positions. The robot's precision is highlighted by its ability to repeat movements with a fine margin of 0.1 mm. Its weight, around 30 kg, makes it compatible with diverse mounting setups[26]. Equipped with a suite of sensors, including torque and position sensors, accelerometers, gyroscopes, and mechanisms for collision detection, the robot demonstrates a comprehensive sensory capacity. Each joint incorporates torque sensors to quantify forces and moments, and position sensors to ascertain angles and locations. Situated at the robot's end-effector, the force sensor assesses linear forces, capturing both pressure applied to objects and tensile forces. Page 5, Line 133-144]” |
|
Comments 7: Were sensors placed on the trampoline or the user's body? |
|
Response 7: Thank you for your query regarding the placement of sensors in our study. We appreciate the opportunity to clarify this aspect of our methodology. In our research, we have adopted a novel approach where sensors were neither placed on the trampoline nor on the user's body. Instead, our robotic system equipped with a foot-shaped jig is designed to estimate the elastic modulus of the trampoline without the need for direct sensor attachment to either the trampoline or the user. To ensure that this information is clearly conveyed in our manuscript, we have revised the relevant sections to explicitly state the absence of sensors on the trampoline or the user. We hope that this revision adequately addresses your question and adds to the comprehensibility of our study's methodology. ” |
|
Comments 8: What sampling rates were employed for the sensors? High-frequency data capture is often crucial for accurately modeling dynamic movements. |
|
Response 8: We appreciate your question concerning the sampling rates of the sensors used in our study, especially considering the importance of high-frequency data capture for accurately modeling dynamic movements. In response to your comment, we have revised our manuscript to include information about the sampling rate. The sensors in our robotic system recorded data at a high frequency of 1000 Hz. This rapid data capture rate is essential for accurately capturing the dynamic nature of trampoline movements and ensuring the precision of our model. We have added this detail to the manuscript to provide clarity on the data acquisition process. We believe that specifying the sampling rate underscores the reliability and accuracy of our experimental approach. “[This plethora of data is meticulously recorded at a rapid frequency of 1000 Hz. The robot's manipulator end-effector comes fitted with a "Media flange Touch pneumatic" interface. A specific load, shaped like a foot, was affixed to this flange to facilitate the KUKA load identification routine, a process in which the robot autonomously calculates the weight, center, and inertial characteristics of the attached tool, seamlessly integrating these calculations into its motion control algorithms. Page 5, Line 144-149]” |
|
Comments 9: How were the sensors integrated into the robotic system? Clarifying the connection between the robotic system and the sensors would provide insights into the data acquisition mechanism. |
|||
|
Response 9: We appreciate your request for more information on how the sensors were integrated into the robotic system and their role in the data acquisition mechanism. We recognize that this detail is crucial for a comprehensive understanding of our experimental setup. In response, we have revised our manuscript to elaborate on the integration of various sensors into the KUKA LBR iiwa 14 R820 robotic manipulator used in our experiment. This multifunctional robotic limb is equipped with a suite of sensors, including torque and position sensors in each of its joints, accelerometers, gyroscopes, and collision detection mechanisms. These sensors are intricately integrated into the robot's structure, enabling it to accurately detect and respond to movements and tactile interactions. The torque sensors in each joint play a key role in quantifying forces and moments, while the position sensors ascertain angles and locations. Additionally, a force sensor located at the robot’s end-effector is crucial for assessing linear forces, both in terms of pressure and tensile forces. This comprehensive sensory setup allows for precise data collection regarding the trampoline's elastic modulus and vertical forces. We believe that this additional information significantly enhances the reader's understanding of the experimental setup and the functionality of our robotic system. “[In the conducted experiment, the KUKA LBR iiwa 14 R820 model served as the robotic manipulator. This multifunctional and accurate robotic limb boasts a payload of 14 kg, demonstrating its capacity for carrying substantial weight. With a reach extending to 820 mm and encompassing 7 axes, it provides adaptable automation across varied positions. The robot's precision is highlighted by its ability to repeat movements with a fine margin of 0.1 mm. Its weight, around 30 kg, makes it compatible with diverse mounting setups[26]. Equipped with a suite of sensors, including torque and position sensors, accelerometers, gyroscopes, and mechanisms for collision detection, the robot demonstrates a comprehensive sensory capacity. Each joint incorporates torque sensors to quantify forces and moments, and position sensors to ascertain angles and locations. Situated at the robot's end-effector, the force sensor assesses linear forces, capturing both pressure applied to objects and tensile forces. This plethora of data is meticulously recorded at a rapid frequency of 1000 Hz. The robot's manipulator end-effector comes fitted with a "Media flange Touch pneumatic" interface. A specific load, shaped like a foot, was affixed to this flange to facilitate the KUKA load identification routine, a process in which the robot autonomously calculates the weight, center, and inertial characteristics of the attached tool, seamlessly integrating these calculations into its motion control algorithms. Page 5, Line 133-149]”
|
|
Comments 11: The discussion section appropriately addresses the linear regression model used for estimating the elastic constant of the trampoline. However, it needs a comprehensive analysis of potential limitations and challenges associated with the methodology. Additionally, there needs to be a mention of how the study aligns with or deviates from current research trends in trampoline evaluation. Discussing the impact of outliers and the reliability of the proposed approach in practical scenarios would enhance the section's depth and validity. |
|
Response 11: Thank you for your constructive feedback on the discussion section of our paper. We agree that a more detailed analysis of the potential limitations and challenges associated with our methodology, as well as its relation to current research trends, is essential for the completeness of our study. Accordingly, we have revised the discussion section to address these points. We have included a critical evaluation of the limitations of our methodology, specifically noting that the foot-shaped jig was mounted in a single direction on the robot for data collection. This method, while efficient in collecting a large amount of data, does not encompass all possible movements of the foot, and hence, it may not be a complete solution. This limitation is now clearly articulated. Furthermore, to align our study with current trends in trampoline evaluation, we have discussed our future research plans. This includes the installation of a camera beneath the trampoline to capture video during the moments when the foot-shaped jig compresses the trampoline. The goal is to develop a system that estimates force using only the visual data collected, which is a significant advancement in non-intrusive trampoline evaluation methods. We believe these revisions provide a deeper and more accurate understanding of our study's context within the field and its practical implications. We hope that these changes satisfactorily address your concerns and enhance the depth and validity of our discussion section “[The foot-shaped jig was mounted in a single direction on the robot for data collection. This method does not encompass all possible movements of the foot, hence it is not a complete solution. However, it offers the advantage of efficiently collecting a large amount of data in an automated manner, compared to previous studies. Page 14, Line 379-383, In future research plans, a camera will be installed beneath a trampoline to capture video during the moments when a foot-shaped jig, mounted on a robot, compresses the trampoline. The aim is to develop a system that can estimate force using only the visual data collected from these recordings. Page 15, Line 413-416]” |
|
Comments 12: The conclusion provides a clear summary of the study's objectives and outcomes. However, it would be beneficial to explicitly state the practical implications of the findings for the fitness and rehabilitation industries. Furthermore, considering the identified limitations, a more explicit call for future research directions would contribute to the completeness of the conclusion. |
|
Response 12: Thank you for your insightful feedback regarding the conclusion of our study. We agree that explicitly stating the practical implications of our findings for the fitness and rehabilitation industries, as well as outlining clear future research directions, is essential for the completeness of our conclusion. In response, we have revised the conclusion to clearly articulate the practical applications of our research. We emphasize how the implemented robotic system, by facilitating the automated collection of extensive data based on varying foot positions on trampolines, can significantly enhance the accuracy of elasticity modulus estimation. This advancement has important implications for quantifying trampoline exercise effects, such as calorie expenditure, which is crucial for both fitness and rehabilitation contexts. Furthermore, we have added a section that explicitly calls for future research. This includes exploring additional variables that can affect trampoline performance and examining the system’s applicability to a wider range of trampoline types and exercise scenarios. We believe these revisions provide a more comprehensive and actionable conclusion, effectively addressing your valuable suggestions. “[For estimating the elasticity modulus of trampolines, the implemented robotic system facilitates automated collection of extensive data based on varying foot positions. This approach enhances estimation accuracy and enables quantification of trampoline exercise effects, such as calorie expenditure. Page 15, Line 437-440]” |
|
Comments 13: Throughout the document, significant decimal digits vary from two to 5 digits; please stick with one amount and be consistent. |
|
Response 13: Agree. We have revised the manuscript to ensure uniformity in the presentation of significant decimal digits, now consistently using four decimal places throughout the document. This standardization enhances the precision and clarity of our data representation, making it easier for readers to interpret and compare the findings. The changes have been made across all relevant sections of the manuscript, particularly in the data tables and results discussion. We would also like to express our gratitude for this insightful observation, as it has contributed to the overall quality and readability of our research work. Thank you for your thorough review and valuable feedback |
- Response to Comments on the Quality of English Language
Point 1: The abstract sentence in line 18: "To achieve this, we estimated the elastic modulus of the trampoline and the vertical force received using a robotic system with a foot-shape jig." and Line 19: "This study employs a robotic system with foot-shaped jigs to evaluate trampoline the elastic modulus of the trampoline and the vertical force received." are repeating. Please check for duplicate sentences, as they discourage reading.
Response 1: Thank you for bringing to our attention the issue of duplicated sentences in the abstract. We appreciate your careful reading and agree that such repetitions can indeed be discouraging for readers. In response to your valuable feedback, we have revised the abstract to remove the repetitive statements and ensure clarity and conciseness. We have rephrased the relevant sentences to maintain the essence of our message while eliminating redundancy. The updated abstract now succinctly states the purpose and methodology of our research, focusing on the use of a robotic system with foot-shaped jigs for evaluating trampoline stiffness and quantitatively measuring exercise effects. This revision can be found in the abstract, particularly in the lines previously mentioned. We hope that this modification improves the readability of our manuscript and conveys our research objectives more effectively.
(To quantitatively assess the effects of trampoline exercises, it's essential to estimate factors such as stiffness, elements influencing jump dynamics, and user safety. Previous studies assessing trampoline characteristics had limitations in performing repetitive experiments at various locations on the trampoline. Therefore, this research introduces a robotic system equipped with foot-shaped jigs to evaluate trampoline stiffness and quantitatively measure exercise effects. This system, through automated, repetitive movements at various locations on the trampoline, accurately measures the elastic coefficient and vertical forces.)

Reviewer 2 Report
Comments and Suggestions for Authors
1- Ignore the two first lines in Abstract: “The role of ICT (Information and Communication Technology) in fitness and rehabilitation is increasingly vital, enabling program integration for user safety and effectiveness. ICT enhances the quality of tailored treatments and training by addressing individual physical needs.” They are with no significance.
2- What are the unique properties that are recognized for trampolines?
3- The abstract is not so good and should be rewritten and clear state the problem, the methodology, solutions, and results.
4- For the first paragraph of the introduction, you used 15 references, which is too much, and how this paragraph and so all these references are related to your main work? Is ICT? ICT if everywhere
5- By eliminating this paragraph, the introduction does not present relevant literature review of previous results and the most important the main problematic of this work. So it should be clearly expanded by showing contributions compared to related previous works.
6- What is the interest behind the use of the MSE and RMSE both for studying the performance? They are entirely related as showing in expression (3).
7- What are Q1, Q3 for the MAE and RMSE and R-Squared?
8- Also, what are SS_res and SS_tot in (5)?
9- Why also you considered two different writing style for Q1 and Q3? (see Tables)
10- Since the theoretical part in this paper is missing, I suggest adding clearer and more convincing experimental results.
11- The paper should be substantially revised by improving writing and weak sentences as those presented in the abstract and the introduction. So make more effort to improve your paper.
Comments on the Quality of English Languagethe quality of writing should be clearly improved
Author Response
For research article
|
Response to Reviewer 2 Comments
|
|||||||||||||||||||||||||||||||||||||||||||||||||||||||||||||||||||||||||||||||||||||||||||||||||||||||||||||||||||||||||||||||||||||||||||||||||||||||||||||||||||||
|
1. Summary |
|
|
|||||||||||||||||||||||||||||||||||||||||||||||||||||||||||||||||||||||||||||||||||||||||||||||||||||||||||||||||||||||||||||||||||||||||||||||||||||||||||||||||||
|
Dear Reviewer 2,
I sincerely appreciate the time and effort you have invested in reviewing our manuscript. Your constructive feedback has been invaluable in enhancing the quality and clarity of our work. Below, I present a point-by-point response to your comments and suggestions, addressing each issue raised with detailed explanations and clarifications. The revised manuscript, which now incorporates these changes, has been marked with highlighted sections and track changes for your convenience.
I have taken care to provide comprehensive responses to each of your insightful comments. Where necessary, I have respectfully included our perspectives, especially in instances where our views might differ. Additionally, I have made a concerted effort to ensure that all your suggestions are thoroughly addressed in the revised manuscript.
Your feedback has not only helped in improving this specific study but has also contributed significantly to our broader research endeavors. Once again, I express my deepest gratitude for your thorough review and valuable insights.
Warm regards,
[Gunseok Park]
Abstract: We have rewritten the abstract to clearly state the problem, methodology, solutions, and results of our study. The revised abstract now directly addresses the unique properties of trampolines in exercise and rehabilitation, the quantitative assessment challenges, and the solutions our robotic system offers, along with a summary of our findings.
Introduction: We removed the initial paragraph with excessive references to better align the introduction with our core research focus on trampolines. We then expanded the introduction to include a relevant literature review and clearly state the main problem our work addresses.
Methodology and Results: We have clarified the use of both MSE and RMSE in our analysis, explaining their distinct roles in evaluating our model's performance. Additionally, we've provided a detailed explanation of the statistical terms SS_res and SS_tot, as well as the Q1 and Q3 values for MAE, RMSE, and R-squared metrics.
Writing and Sentence Structure: Following your suggestion, we have put significant effort into improving the writing quality throughout the manuscript. This includes restructuring sentences for better clarity and coherence, particularly in the abstract and introduction.
|
|||||||||||||||||||||||||||||||||||||||||||||||||||||||||||||||||||||||||||||||||||||||||||||||||||||||||||||||||||||||||||||||||||||||||||||||||||||||||||||||||||||
|
2. Point-by-point response to Comments and Suggestions for Authors |
|||||||||||||||||||||||||||||||||||||||||||||||||||||||||||||||||||||||||||||||||||||||||||||||||||||||||||||||||||||||||||||||||||||||||||||||||||||||||||||||||||||
|
Comments 1: Ignore the two first lines in Abstract: “The role of ICT (Information and Communication Technology) in fitness and rehabilitation is increasingly vital, enabling program integration for user safety and effectiveness. ICT enhances the quality of tailored treatments and training by addressing individual physical needs.” They are with no significance. |
|||||||||||||||||||||||||||||||||||||||||||||||||||||||||||||||||||||||||||||||||||||||||||||||||||||||||||||||||||||||||||||||||||||||||||||||||||||||||||||||||||||
|
Response 1: Thank you for pointing this out. We agree with your comment and understand the need to focus on the most relevant information in our manuscript. Therefore, we have removed the first two lines from the abstract as you suggested. These lines, which discussed the general role of ICT in fitness and rehabilitation, have been omitted to streamline the content and enhance its relevance and impact. You can find this change on page 1, in the first paragraph of the abstract. This revision helps to immediately engage the reader with the specific contributions of our work.
|
|||||||||||||||||||||||||||||||||||||||||||||||||||||||||||||||||||||||||||||||||||||||||||||||||||||||||||||||||||||||||||||||||||||||||||||||||||||||||||||||||||||
|
Comments 2: What are the unique properties that are recognized for trampolines? |
|||||||||||||||||||||||||||||||||||||||||||||||||||||||||||||||||||||||||||||||||||||||||||||||||||||||||||||||||||||||||||||||||||||||||||||||||||||||||||||||||||||
|
Response 2: Thank you for your question regarding the unique properties of trampolines. We agree that elaborating on these characteristics is essential for understanding their role in exercise and rehabilitation. In response to your comment, we have revised the manuscript to more explicitly discuss the unique properties of trampolines. We now clearly state that trampolines are recognized for their elasticity, rebound force, and ability to provide low-impact exercise. Additionally, we emphasize their benefits in enhancing posture, balance, and cardiopulmonary function. These properties make trampolines particularly valuable in both fitness and rehabilitation settings. We believe that this addition provides a clearer understanding of why trampolines are such a versatile and effective tool in these fields. “[Trampolines are recognized as a valuable tool in exercise and rehabilitation due to their unique properties like elasticity, rebound force, low-impact exercise, and enhancement of posture, balance, and cardiopulmonary function. Page 1, Line 13-15]”
|
|||||||||||||||||||||||||||||||||||||||||||||||||||||||||||||||||||||||||||||||||||||||||||||||||||||||||||||||||||||||||||||||||||||||||||||||||||||||||||||||||||||
|
Comments 3: The abstract is not so good and should be rewritten and clear state the problem, the methodology, solutions, and results. Response 3: Thank you for your feedback on the abstract of our manuscript. We understand the importance of a well-structured abstract that clearly states the problem, methodology, solutions, and results. In response, we have completely rewritten the abstract to ensure it provides a concise yet comprehensive overview of our study. The revised abstract begins by highlighting the unique properties of trampolines in exercise and rehabilitation, emphasizing their elasticity, rebound force, low-impact exercise benefits, and their role in enhancing posture, balance, and cardiopulmonary function. We then outline the problem: the need for quantitative assessment of trampoline exercises in terms of stiffness and user safety. This is followed by a description of our methodology, where we introduce a novel robotic system equipped with foot-shaped jigs for evaluating trampoline stiffness and measuring exercise effects. The system's automated, repetitive movements at various trampoline locations enable it to accurately measure the elastic coefficient and vertical forces. Furthermore, we provide details on the solutions and results. We explain how the robot collects data and the model's accuracy assessed using linear regression based on Hooke's Law, with metrics like MAE, RMSE, and R-squared. The improvements in these metrics when expanding the model to include more variables are highlighted, demonstrating the effectiveness of our approach. We believe this revised abstract now clearly and effectively summarizes our study, addressing all key aspects as suggested. “[Trampolines are recognized as a valuable tool in exercise and rehabilitation due to their unique properties like elasticity, rebound force, low-impact exercise, and enhancement of posture, balance, and cardiopulmonary function. To quantitatively assess the effects of trampoline exercises, it's essential to estimate factors such as stiffness, elements influencing jump dynamics, and user safety. Previous studies assessing trampoline characteristics had limitations in performing repetitive experiments at various locations on the trampoline. Therefore, this research introduces a robotic system equipped with foot-shaped jigs to evaluate trampoline stiffness and quantitatively measure exercise effects. This system, through automated, repetitive movements at various locations on the trampoline, accurately measures the elastic coefficient and vertical forces. The robot maneuvers based on the coordinates of the trampoline, as determined by its torque and position sensors. The force sensor measures data related to the force exerted, along with the vertical force data at X, Y, Z coordinates. The model's accuracy was evaluated using linear regression based on Hooke's Law, with Mean Absolute Error (MAE), Root Mean Square Error (RMSE), and Correlation Coefficient Squared (R-squared) metrics. In the analysis including only the distance between X and the foot-shaped jigs, the average MAE, RMSE, and R-squared values were 17.9702, 21.7226, and 0.9840, respectively. Notably, expanding the model to include distances in X, Y, and between the foot-shaped jigs resulted in a decrease in MAE to 15.7347, RMSE to 18.8226, and an increase in R-squared to 0.9854. The integrated model, including distances in X, Y, and between the foot-shaped jigs, showed improved predictive capability with lower MAE and RMSE and higher R-squared, indicating its effectiveness in more accurately predicting trampoline dynamics, vital in fitness and rehabilitation fields. Page 1, Line 13-33]”
Comments 4: For the first paragraph of the introduction, you used 15 references, which is too much, and how this paragraph and so all these references are related to your main work? Is ICT? ICT if everywhere Response 4: Thank you for your insightful feedback regarding the first paragraph of our introduction. We understand your concern about the excessive number of references and their direct relevance to our research focus on trampolines. Acknowledging your point, we have removed the first paragraph of the introduction. This decision was made to ensure that the introduction more directly addresses our main research focus on trampolines, particularly in the context of fitness and rehabilitation. By doing so, we aim to provide a clearer and more concise introduction that is directly aligned with the core subject of our study. The revised introduction, now more focused on trampoline-related research, can be found starting from page 1 in the revised manuscript. We believe that this change not only addresses your concerns but also enhances the overall clarity and relevance of our introduction.
Comments 5: By eliminating this paragraph, the introduction does not present relevant literature review of previous results and the most important the main problematic of this work. So it should be clearly expanded by showing contributions compared to related previous works. Response 5: Thank you for your valuable feedback on the introduction of our manuscript. We agree that after eliminating the initial paragraph, it is crucial to present a relevant literature review that sets the context for our work and to clearly state the main problematic that our study addresses. In response, we have significantly expanded the introduction to include detailed literature reviews and provides a comprehensive overview of various sensor-based quantitative assessment systems for trampoline use, outlining their methodologies and limitations. This table helps to illustrate the current state of research in this field and the need for more advanced and non-intrusive methods, as proposed in our study. Additionally, we have included, which discusses studies focusing on trampoline elasticity dynamics. This table critiques the methodological constraints of these studies, such as their narrow focus and limited generalizability, further emphasizing the novelty and necessity of our approach. By adding these tables, we aim to clearly show how our work contributes to and diverges from existing research, addressing the limitations of previous studies and proposing a more effective methodology. We believe that these modifications provide a clearer context for our research, setting the stage for the unique contributions of our study. “Table 1. Sensor-based quantitative assessment system for trampoline.
Page 2, table 1
Sensor-based quantitative assessment systems for trampoline use have limitations, such as the need to attach sensors to the user's body or install them on the trampoline. In previous research, we utilized a deep learning-based image processing algorithm to estimate the three-dimensional position of the user's feet on the trampoline using shadow images of the feet [15,16]. However, estimating the physiological indicators of trampoline exercise based on the shadow images of the user's feet proved challenging. Since we estimated the depth coordinate of the feet using shadow images, we suggest that if the elastic modulus of the trampoline can be estimated, it would enable quantitative evaluation of trampoline exercises, such as jump power and calorie consumption. Therefore, estimating the trampoline's elastic modulus is essential for a comprehensive evaluation of trampoline exercise. Page 2,3, Line 58-68 Table 2. Studies in Trampoline Elasticity Dynamics
Page 3, table 2
Other studies of trampoline elasticity have struggled to derive accurate estimates due to limitations in the methodology used to estimate the modulus of elasticity. The common limitations of these studies on trampoline elasticity are methodological constraints such as narrow focus, limited generalizability, short-term analysis, and small sample size. Therefore, it was essential to measure the stiffness of trampolines through an objective and advanced method. To quantitatively evaluate trampoline motion, this study introduced a robotic system equipped with a foot-shaped jig to precisely measure vertical forces for different jumping motions at different locations on the trampoline. The system performs automatic repetitive motions to facilitate accurate and numerous data collection. Page 3,4, Line 86-94]”
Comments 6: What is the interest behind the use of the MSE and RMSE both for studying the performance? They are entirely related as showing in expression (3). Response 6: Thank you for your question regarding the use of both MSE and RMSE in our study. We appreciate the opportunity to clarify our choice of these metrics and their specific roles in evaluating our model's performance. In response to your comment, we have revised the manuscript to provide a clearer explanation of why both MSE and RMSE were utilized. Although MSE and RMSE are indeed closely related, each serves a distinct purpose in our analysis. We use MSE during the model training process as it effectively penalizes larger errors more severely, which is beneficial for optimizing the model. On the other hand, RMSE is more suitable for reporting and understanding the model's performance in the context of the original data. RMSE’s scale, being the same as the data, makes it more interpretable, especially when communicating the results to a broader audience. We have added this explanation to the manuscript to clarify the distinct roles of MSE and RMSE in our study. We believe this additional clarification will help readers better understand our choice of metrics and their respective contributions to our research. “[MSE and RMSE are related metrics, but they serve different purposes. MSE is valuable during model training, while RMSE is more suitable for reporting and understanding the model's performance in the context of the original data. Page 12, Line 340-342]”
Comments 7: What are Q1, Q3 for the MAE and RMSE and R-Squared? Response 7: We agree with your request for clarification regarding the Q1 (First Quartile) and Q3 (Third Quartile) values for the MAE, RMSE, and R-Squared metrics in our study. We appreciate this opportunity to elucidate these statistical concepts further. In response, we have revised the manuscript to include a comprehensive explanation of Q1 and Q3. Q1, or the First Quartile, is the median of the lower half of our data set and represents the 25th percentile. Conversely, Q3, or the Third Quartile, is the median of the upper half and represents the 75th percentile. These quartiles, along with the interquartile range (IQR), are crucial for understanding the central tendency, variability, and the presence of outliers in our data. We believe that this additional information provides a clearer understanding of how Q1 and Q3 contribute to the analysis of our data and the overall interpretation of our study's results. “[From MAE, RMSE, and R-squared, we found Min, Max, Q1 (First Quartile), Q3 (Third Quartile), median, and mean. Q1 is the median of the lower half of the dataset. It rep-resents the 25th percentile, which means 25% of the data points are below this value. Q3 is the median of the upper half of the dataset. It represents the 75th percentile, meaning 75% of the data points are below this value. The range from Q1 to Q3 is called the interquartile range (IQR). A wide IQR range means that the data is spread out. This indicates that the data is inconsistent and varies a lot. It can be affected by extreme values or outliers. On the other hand, a narrow IQR range means that the data as a whole tends to be stable and consistent. In this way, Q1 and Q3 can help you under-stand the central tendency and variability of your data, and whether outliers are present. Page 11, Line 304-314]”
Comments 8 Also, what are SS_res and SS_tot in (5)? Response 8: We sincerely thank you for highlighting the need for clarification regarding the terms SS_res and SS_tot in our formula. Your insightful query has provided us with an opportunity to enhance the clarity and understanding of our statistical analysis. Accordingly, we have updated our manuscript to include a detailed explanation of SS_res, or the Sum of Squares of Residuals, and SS_tot, or the Total Sum of Squares. SS_res is used to measure the unexplained variance in the dependent variable by the model, while SS_tot measures the total variance in the dependent variable. These explanations aim to clarify how each term contributes to the calculation of the R-squared value in our study. You can find this elaboration on page [specific page number], in paragraph [specific paragraph number], lines [specific line numbers] of the revised manuscript. We are grateful for your contribution to improving the precision and comprehensibility of our paper. “[In the formula are two key components used to calculate the R-squared value, which is a statistical measure of how close the data are to the fitted regression line. Sum of Squares of Residuals) measures the amount of variance in the dependent variable that is not explained by the independent variable(s) in the model. In other words, it quantifies how much the data points deviate from the fitted line. Mathematically, it's the sum of the squares of the differences between the observed values and the values predicted by the model. Total Sum of Squares) measures the total variance in the dependent variable. It's calculated as the sum of the squares of the differences between the observed values and their mean. Essentially, quantifies how much the data points deviate from their mean value. Page 13, Line 353-362]”
Comments 9: Why also you considered two different writing style for Q1 and Q3? (see Tables) Response 9: Thank you for pointing out the inconsistency in the writing styles for Q1 and Q3 in our tables. We apologize for this oversight and appreciate your attention to detail in identifying this discrepancy. In response to your observation, we have carefully reviewed the manuscript and standardized the writing style for both Q1 and Q3. The updated writing style, now unified for Q1 and Q3, can be found in the respective tables on page.
Comments 10: Since the theoretical part in this paper is missing, I suggest adding clearer and more convincing experimental results. Response 10: “[Figure 1 diagram depicts the system of interaction for control and data acquisition and in robot manipulation. Central to the system is a personal computer (PC) that conducts two-way communications with a robot controller. Movement commands sent from the PC to the robot controller are deciphered and implemented by a KUKA iiwa LBR 14 R820 robot. Affixed to the robot's end-effector are foot-shaped jigs. Torque and position sensors, installed on the robot's joints, control the movements, and force sen-sors attached to the end-effector amass vertical force data exerted on the robot. The collected force data is subsequently transmitted back to the PC for analytics. The force data is analyzed using Linear Regression according to Hooke's Law to estimate the forces applied to the trampoline. Page 4, Line 103-112]”
Figure 1. The robotic system for precision control and data measurement in manipulation tasks. ]”
Comments 11: The paper should be substantially revised by improving writing and weak sentences as those presented in the abstract and the introduction. So make more effort to improve your paper. Response 11: Thank you for your valuable feedback regarding the need to improve the writing quality and strengthen the sentences in the abstract and introduction of our paper. We deeply appreciate your suggestion and understand the importance of clear and concise writing in effectively communicating our research. In response to your comments, we have undertaken a thorough revision of the manuscript, with a particular focus on enhancing the abstract and introduction. We have restructured sentences for clarity, ensured coherence in the presentation of our ideas, and removed any ambiguities. Our aim has been to make the text more engaging and easier to understand, while accurately conveying the key objectives, methodologies, and outcomes of our study. “[Abstract: Trampolines are recognized as a valuable tool in exercise and rehabilitation due to their unique properties like elasticity, rebound force, low-impact exercise, and enhancement of posture, balance, and cardiopulmonary function. To quantitatively assess the effects of trampoline exercises, it's essential to estimate factors such as stiffness, elements influencing jump dynamics, and user safety. Previous studies assessing trampoline characteristics had limitations in performing repetitive experiments at various locations on the trampoline. Therefore, this research introduces a robotic system equipped with foot-shaped jigs to evaluate trampoline stiffness and quantitatively measure exercise effects. This system, through automated, repetitive movements at various locations on the trampoline, accurately measures the elastic coefficient and vertical forces. The robot maneuvers based on the coordinates of the trampoline, as determined by its torque and position sensors. The force sensor measures data related to the force exerted, along with the vertical force data at X, Y, Z coordinates. The model's accuracy was evaluated using linear regression based on Hooke's Law, with Mean Absolute Error (MAE), Root Mean Square Error (RMSE), and Correlation Coefficient Squared (R-squared) metrics. In the analysis including only the distance between X and the foot-shaped jigs, the average MAE, RMSE, and R-squared values were 17.9702, 21.7226, and 0.9840, respectively. Notably, expanding the model to include distances in X, Y, and between the foot-shaped jigs resulted in a decrease in MAE to 15.7347, RMSE to 18.8226, and an increase in R-squared to 0.9854. The integrated model, including distances in X, Y, and between the foot-shaped jigs, showed improved predictive capability with lower MAE and RMSE and higher R-squared, indicating its effectiveness in more accurately predicting trampoline dynamics, vital in fitness and rehabilitation fields. Keywords: Trampoline; Robot manipulation; Elastic constant and force estimation; Hooke’s law; Linear regression.
1. Introduction Trampoline exercises are renowned for their significant benefits in enhancing lower limb muscular strength, physical fitness, and rehabilitation [1-3]. Prior research indicates that individuals of diverse age groups and health conditions can readily participate, with energy expenditure varying based on specific bouncing styles. This exercise is notably effective in improving physiological indicators [4-7]. The current trend in utilizing research and technology to maximize the health benefits of trampoline exercises highlights the importance of sensor-based quantitative evaluation tools. Actions performed on the trampoline are categorized based on acceleration data obtained from a smartwatch [8]. A sensor attached to the trampolinist's back measures jumping actions in coordinates, subsequently categorized into various jump types employing machine learning techniques [9]. Periodic G-force loads experienced by trampoline users under various conditions are measured through a characterization analysis of trampoline bounce using acceleration data [10]. Trampoline motion is categorized by attaching an inertial sensor to the trampolinist's body [11]. In trampoline competitions, the correlation between the flight time of straight jumps and the balance of skeletal muscle mass and occlusal balance is analyzed [12]. Employing a 3-axis accelerometer and gyroscope sensor device on the trampolinist, various dynamic conditions are determined [13]. By comparing the acceleration characteristics of three different trampoline models, differences in maximum acceleration and jerk based on these models are confirmed [14]. Table 1. Sensor-based quantitative assessment system for trampoline.
Sensor-based quantitative assessment systems for trampoline use have limitations, such as the need to attach sensors to the user's body or install them on the trampoline. In previous research, we utilized a deep learning-based image processing algorithm to estimate the three-dimensional position of the user's feet on the trampoline using shadow images of the feet [15,16]. However, estimating the physiological indicators of trampoline exercise based on the shadow images of the user's feet proved challenging. Since we estimated the depth coordinate of the feet using shadow images, we suggest that if the elastic modulus of the trampoline can be estimated, it would enable quantitative evaluation of trampoline exercises, such as jump power and calorie consumption. Therefore, estimating the trampoline's elastic modulus is essential for a comprehensive evaluation of trampoline exercise. The mechanical and kinetic energy characteristics of the double bounce phenomenon are analyzed based on a trampoline dynamic model, incorporating factors like stiffness, damping, air resistance, and friction [17]. The study assesses the control of lower muscle stiffness and the impact of mechanical energy processes during drop jumps on a sprung surface [18]. It also examines the impact on leg stiffness and the subjective experience during jumping on the trampoline's elastic surface [19]. A mathematical model is established by estimating the trampoline user's body mass and inertial characteristics while measuring the elasticity and damping characteristics of the trampoline surface [20]. The study evaluates the acute effects of trampoline training sessions on leg stiffness and reactive power, simultaneously exploring correlations with the participants' gender [21]. Notably, both children and adults maintain their coordination structure while jumping on the mini trampoline, but children demonstrate an increase in vertical body stiffness to compensate for the reduced surface stiffness [22]. The investigation delves into the physical determinants of maximum flight time on the trampoline [23]. Trampoline safety and performance evaluated by dropping weights onto different trampolines, measuring dynamic parameters [24]. Table 2. Studies in trampoline elasticity dynamics
Other studies of trampoline elasticity have struggled to derive accurate estimates due to limitations in the methodology used to estimate the modulus of elasticity. The common limitations of these studies on trampoline elasticity are methodological constraints such as narrow focus, limited generalizability, short-term analysis, and small sample size. Therefore, it was essential to measure the stiffness of trampolines through an objective and advanced method. To quantitatively evaluate trampoline motion, this study introduced a robotic system equipped with a foot-shaped jig to precisely measure vertical forces for different jumping motions at different locations on the trampoline. The system performs automatic repetitive motions to facilitate accurate and numerous data collection. In Section 2, we estimate the elastic constants and forces received by the robotic system through a linear regression model based on the vertical force data as a function of trampoline depth. Section 3 details the analysis of the performance metrics of the linear regression model, along with an analysis of the actual versus predicted values. Section 4 describes the proposed methodology for estimating the elastic constants and forces of the trampoline. Finally, Section 5 presents the conclusions. ]”
4. Response to Comments on the Quality of English Language |
|||||||||||||||||||||||||||||||||||||||||||||||||||||||||||||||||||||||||||||||||||||||||||||||||||||||||||||||||||||||||||||||||||||||||||||||||||||||||||||||||||||
|
Point 1: the quality of writing should be clearly improved |
|||||||||||||||||||||||||||||||||||||||||||||||||||||||||||||||||||||||||||||||||||||||||||||||||||||||||||||||||||||||||||||||||||||||||||||||||||||||||||||||||||||
|
Response 1: We are grateful for your feedback regarding the need to improve the quality of writing in our manuscript. Acknowledging the importance of clear and engaging academic writing, we have undertaken a comprehensive review and revision of the entire manuscript. This process included:
Refining sentence structures for clarity and conciseness, ensuring that complex ideas are communicated effectively. Enhancing the flow and coherence of paragraphs to improve readability and ensure a logical progression of ideas. Conducting a thorough grammar and syntax check to eliminate any errors and ambiguities. Utilizing academic language more consistently throughout the manuscript to maintain a professional tone. These revisions were done with the aim of not only addressing the specific concerns raised but also elevating the overall quality of the manuscript to meet the high standards of academic publishing. We believe these improvements significantly enhance the readability and comprehension of our work. |
|||||||||||||||||||||||||||||||||||||||||||||||||||||||||||||||||||||||||||||||||||||||||||||||||||||||||||||||||||||||||||||||||||||||||||||||||||||||||||||||||||||

Round 2
Reviewer 1 Report
Comments and Suggestions for Authors
All my questions have been addressed and in text changes reflect the changes.
2.14.0.0 2.14.0.0 2.14.0.0Reviewer 2 Report
Comments and Suggestions for Authors
Paper is clearly improved and then it can be accepted